# Effects of Multi-Omics Characteristics on Identification of Driver Genes Using Machine Learning Algorithms

**DOI:** 10.3390/genes13050716

**Published:** 2022-04-19

**Authors:** Feng Li, Xin Chu, Lingyun Dai, Juan Wang, Jinxing Liu, Junliang Shang

**Affiliations:** School of Computer Science, Qufu Normal University, Rizhao 276826, China; lifeng_10_28@163.com (F.L.); chuxinqf@163.com (X.C.); dailingyun_1@163.com (L.D.); wangjuansdu@163.com (J.W.); sdcavell@qfnu.edu.cn (J.L.)

**Keywords:** pan-cancer, multi-omics, driver gene, machine learning, Kullback–Leibler divergence

## Abstract

Cancer is a complex disease caused by genomic and epigenetic alterations; hence, identifying meaningful cancer drivers is an important and challenging task. Most studies have detected cancer drivers with mutated traits, while few studies consider multiple omics characteristics as important factors. In this study, we present a framework to analyze the effects of multi-omics characteristics on the identification of driver genes. We utilize four machine learning algorithms within this framework to detect cancer driver genes in pan-cancer data, including 75 characteristics among 19,636 genes. The 75 features are divided into four types and analyzed using Kullback–Leibler divergence based on CGC genes and non-CGC genes. We detect cancer driver genes in two different ways. One is to detect driver genes from a single feature type, while the other is from the top N features. The first analysis denotes that the mutational features are the best characteristics. The second analysis reveals that the top 45 features are the most effective feature combinations and superior to the mutational features. The top 45 features not only contain mutational features but also three other types of features. Therefore, our study extends the detection of cancer driver genes and provides a more comprehensive understanding of cancer mechanisms.

## 1. Introduction

Cancer is one of the most difficult diseases to treat and one of the most dangerous to human health [1]. It is a complex disease caused by different kinds of genetic alterations, which can disrupt cell proliferation and death during a person’s lifetime [2,3]. Recent developments in the field of next-generation sequencing (NGS) [4] offer unprecedented opportunities to better describe the molecular characteristics of human cancers. The Cancer Genome Atlas (TCGA) [5] and the International Cancer Genome Consortium (ICGC) [6] have amassed and analyzed a substantial amount of cancer genomic data [7]. Genes with mutations or copy number alterations that accelerate cancer evolution are called cancer drivers [3]. Multiple different driver genes work together to gradually transform normal cells into invasive and metastatic tumors [8]. Mutational features are unique combinations of mutation types caused by distinct mutagenesis processes. Like deoxyribonucleic acid (DNA) replication infidelity, DNA enzymatic editing results in mutational signatures, which are distinct combinations of mutation types. Epigenetics most often involves changes that affect gene activity and expression. External or environmental influences may affect cellular and physiological features, or they may be a normal aspect of development [9,10,11]. Some critical epigenetic modifications often play an important role in cancer and affect gene activity and expression to promote various metabolic, autoimmune, and neurological diseases [12,13]. For example, H3 lysine 4 (H3K4me3) and 5′—C—phosphate—G—3′ (CpG) methylation alteration are related to transcription elongation, enhancer activity, and repression of tumor suppressors [14]. Genomic features include the maximum number of protein–protein interactions, biological principle types of cells, and post-translational modification (PTM) [15].

Therefore, it can show a more comprehensive view to identify driver genes by considering both genomics and epigenomics information. Most methods for detecting driver genes are based on a genomic mutation dataset, while some algorithms use both somatic mutation data and copy number alterations. Positive selection is a major evolutionary force in cancer, resulting in the accumulation of driving mutations in critical genes that promote tumor growth [16]. This is to distinguish driver mutations, providing fitness benefits to cells under selective pressure, from passenger mutations [17]. Tokheim et al. looked back at eight major algorithms, and Bailey et al. integrated 26 computational tools in a pan-cancer mutation study [17]. Tumor suppressor and Oncogenes Explorer (TUSON) [18] and 20/20+ machine learning methods [19] are the two main algorithms that can distinguish between tumor suppressor genes (TSGs) and oncogenes (OGs) encoding proteins based on differences in the unique patterns of the mutation characteristics. However, many cancer driver genes will not be discovered because of their high heterogeneity in populations [20,21]. Therefore, the efficient use of epigenomic data and genomic data can improve the prediction of cancer-driving genes [22]. Based on features integrating protein–protein interactions (PPIs) at the genomic and mutational level, it is possible to identify whether a driver or a passenger is a somatic mutation [15].

The main aim of this study is to present a comprehensive analysis of multi-omics characteristics, which are more likely to contribute to the identification of cancer driver genes. We provide a framework to analyze the influence of multiple omics features on driver gene identification. In this framework, four machine learning [23] algorithms are used to detect cancer driver genes in pan-cancer data, which contain 75 characteristics among 19,636 genes [22]. We divide these 75 features into four types and analyze them using Kullback–Leibler divergence [24] based on Cancer Gene Census (CGC) genes and non-CGC genes. Then, we detect cancer driver genes in two different ways. One is to detect driver genes from a single feature type to discuss which type of feature has the best characteristics. Meanwhile, the other one is to detect driver genes from the top N features for discussing which combinations of features are the most effective. We also compare the framework with other methods and analyze the driver genes detected by four machine learning algorithms.

## 2. Materials and Methods

### 2.1. Data Resources

To analyze the effects of multi-omics characteristics on the identification of driver genes, we apply this framework to analyze pan-cancer data for 75 features found in 19,636 genes from 33 cancer types [17,22], which is derived from the TCGA website (https://portal.gdc.cancer.gov/ (accessed on 1 August 2021)) and Catalogue of Somatic Mutations in Cancer (COSMIC) [25]. Combining these two datasets helps to increase the mutation information of genes with less common mutations.

These characteristics are classified into four broad categories [22]: (i) 33 mutational features from two commonly used cancer driver gene prediction algorithms, TUSON and 20/20+ [19], and Genome Aggregation Database [2]. A total of 28 of these 33 features were compiled from the mutation data of patient samples by TCGA [26] and COSMIC [25]; (ii) 12 genomic features, including 3 from 20/20+ and 9 features (e.g., gene lengths and characteristics related to genome evolution) that haven’t been used to predict cancer driver genes before [27]; (iii) 27 epigenetic features, including histone modifications from the ENCODE project [28], super-enhancer percentages from the dbSUPER database, as well as promoter and gene-body methylation properties from the COSMIC database [29]; and (iv) 3 phenotypic features, including CRISPR-screening data from the DepMap project, Variant Effect Scoring Tool (VEST) scores from 20/20+, and gene expression Z scores from TCGA.

In general, supervised machine learning requires labeled genes to train a classifier. We also downloaded a list of 723 CGC genes from the COSMIC database as known driver genes [30,31], which is the benchmark data in this work.

The framework of analyzing the influence of multiple omics features on driver gene identification is shown in Figure 1. Firstly, these 75 features are analyzed using KL divergence based on CGC genes and non-CGC genes. Then, we utilize four machine learning algorithms including random forest, logistic regression, XGBoost, and neural networks, to predict cancer driver genes. This is because the 75 features are divided into four types based on the known literature, including 12 genomic features, 33 mutational features, 27 epigenetic features, and 3 phenotypic features. Thus, we detect cancer driver genes in two different ways. One way is to detect driver genes from a single feature type, while the other way is to detect driver genes from the top N features. At last, we analyze the driver genes detected from different features.

### 2.2. Kullback–Leibler Divergence

We use Kullback–Leibler divergence (KL divergence) [24] to analyze these 75 features based on CGC genes and non-CGC genes. KL divergence is also known as relative entropy, information divergence, or information gain. Solomon Kullback and Richard Leibler introduced KL divergence as the directed divergence between two distributions [24]. Consider two probability distributions P and Q. In this work, KL divergence measures the importance of each feature between cancer genes and other non-cancer genes.

For two discrete probability distributions P and Q, defined on the same probability space *x*, the relative entropy from Q to P is defined to be:(1)$$KL\left(P||Q\right)=\sum P\left(x\right)\log\frac{P\left(x\right)}{Q\left(x\right)}$$
where the average of $KL\left(P||Q\right)$ and $KL\left(Q||P\right)$ is the final *KL* distance.

### 2.3. Detection Method

Four supervised learning models are trained to detect driver genes such as random fores, logistic regression [32], XGBoost [33], and neural networks [34].

#### 2.3.1. Logistic Regression

Logistic regression [35] is a classification algorithm and is familiar with linear regression. Logistic regression has a general form y=ax+b, and the value range y is random. By entering the result y into the sigmoid function of a nonlinear transformation, y can be taken into account as a probability value with [0, 1]. If we set the probability threshold to 0.5, y greater than 0.5 can be regarded as a driver gene. Less than 0.5 is regarded as a non-driver gene. Then, all genes can be classified.

The kernel function of logistic regression is as follows:(2)$$h_{\theta }\left(x\right)=g\left(\theta^{T}x\right)=\frac{1}{1+e^{-\theta^{Tx}}}$$
where h is the prediction function, x stands for the genetic trait, and θ is the parameter of each feature.

#### 2.3.2. Random Forest

Random forest algorithm [36] can randomly build a decision tree. The random forest belongs to the bagging algorithm in ensemble learning. Therefore, random forest is characterized by the weak generalization ability of decision trees. After obtaining the forest, there is a new input as the gene feature, and each decision tree in the forest needs to be judged separately. We use the objective function of random forest to get the top value, which may be the potential cancer driver gene.

Kernel function of random forest [37] is:(3)$$g\left(t\right)=\frac{c\left(t\right)-c\left(T_{t}\right)}{\left|T_{t}\right|-1}$$
where $T_{t}$ represents the subtree with t as the root node, $c\left(T_{t}\right)$ is the prediction error of the training data set, and $\left|T_{t}\right|$ is the number of leaf nodes of $T_{t}$.

#### 2.3.3. XGBoost

XGBoost [33] is an essentially gradient boosting decision, which has maximum speed and efficiency. Trees are constantly added, while a new function is being learned to fit the residuals of the last prediction. Each tree will fall into a leaf node based on the properties of the driver gene, and each leaf node correlates to a score. Finally, simply add the scores of each tree to obtain the predicted value of the gene based on the threshold of the objective function.

The objective function of XGBoost is as follows:(4)$$L\left(\phi \right)=\sum_{i}l\left(y_{i}^{’}-y_{i}\right)+\sum_{k}\left(\gamma T+\frac{1}{2}\lambda \sum_{j=1}^{T}w_{j}^{2}\right)$$
where T is the number of leaves in the tree, y is the label, l is the module square of the score, and w is the leaf node in the tree.

#### 2.3.4. Neural Network

The neural network used in this work is an artificial neural network. The artificial neural network [34] trains the multi-layer feedforward network through the error backpropagation algorithm. The error gradient descent method is used to ensure the error signal. Under the minimum premise, modify the weight and threshold of each layer of neurons. Adjusting the training function and transfer function of the deep neural network can realize complex nonlinear mapping problems. We use this property of neural networks to predict potential cancer genes more accurately.

We used three layers, which include the input layer, output layer, and hidden layer. When a gene feature goes from the input layer, to the hidden layer, and then to the output layer, the first neural network is to substitute the gene feature value into the Relu function, that is:(5)$$f\left(x\right)=\max\left(0,x\right)$$

The output of the i-th node in the hidden layer is:(6)$$r_{i}=f\left(\sum_{j=1}^{n}w_{ij}p_{j}+\theta_{i}\right)$$
where $\theta_{i}$ is the threshold of hidden layer nodes, p stands for neuron, and $w_{ij}$ is the weight between node i and node j.

### 2.4. Five-Fold Cross-Validation

We use fivefold cross-validation to process the 75 features to obtain a reliable and stable supervised learning model. To address the challenges of the imbalance of cancer gene datasets, we use undersampling and five-fold cross-validation based on most classes. In this work, we replace oversampling with undersampling. Sampling is not used because it is prone to over-fitting. Among 19,636 genes, 698 genes are labeled as cancer genes. The remaining 18,938 non-CGC genes are labeled as non-cancer genes. A total of 80% of the genes are randomly selected as the training set and the remaining 20% as the test set. There are 15,150 non-cancer genes and 558 cancer genes in the training set. The test set has 3788 non-cancer genes and 140 cancer genes. The average of the results of 100 runs is the final result.

### 2.5. Performance Evaluation

We use the CGC genes as an approximate benchmark for known driver genes. For comparison, we use four indicators to evaluate the performance. The four indicators are accuracy, recall, precision, and F1-score. The four indicators are introduced below.

*Accuracy* predicts the correct gene number/total gene number, and its formula is as follows:(7)$$Accuracy=\frac{TP+TN}{TP+FP+TN+FN}$$

*Precision* is the proportion of genes that are positive in all genes that are predicted to be positive, and its formula is as follows:(8)$$Precision=\frac{TP}{TP+FP}$$

*Recall* is the proportion that multiple positives are classified as positives, and its formula is as follows:(9)$$Recall=\frac{TP}{TP+FN}$$

*Specificity* is the correct proportion of all negative genes, and its formula is as follows:(10)$$Specificity=\frac{TN}{N}$$

In Formulas (7)–(10), TP is truly positive, FP is false positive, FN is false negative, and TN is a true negative. N is short for negative, which is the sum of FP and TN.

*F1-score* is a comprehensive evaluation index, which is the harmonic mean of precision and recall. Its formula is as follows:(11)$$F1-score=2\frac{Precision\times Recall}{Precision+Recall}$$

## 3. Results

### 3.1. Feature Importance of KL Divergence

We use KL divergence to measure the importance of each feature between cancer genes and non-cancer genes. The importance of all 75 features is sorted by KL divergence, which is shown in Figure 2. We describe the feature set in more detail (Appendix A) [22]. All importance levels are between 0 and 0.6, where 0 is the least important feature. The higher the KL divergence value, the more important the feature. We can see that the performance of the four types of features is quite different. The mutation features have the highest score. The epigenetic feature also performed well; the Height_of_H4K20me1_peaks feature has the highest score. The genomic feature is also good, only inferior to the epigenetic feature. Phenotype features are not as obvious as the other three, but they also play a certain role.

In terms of individual feature categories, in the mutation features, the ranking of log Total N LoFlog, Total N missense mutations, and missense mutations/KB plays an important role. For information on the density of various types of mutations inside a gene, only the coding DNA sequence (CDS) of each gene is considered. In epigenetics features, the percentage of broad H4K20me1 peaks in ENCODE samples, H3K4me1 peak length, and H3K4me2 peak length play an important role. In genomics features, residual variation scores (RVIS) percentile, non-coding genomic evolutionary rate profiling score, and exon conservation score play an important role. It is based on the average phastCons score, and the maximum transcript of genes is also calculated by CRAVAT. As shown in Figure 2, sequencing of gene features according to KL measurement show that the top ten are mutation features.

### 3.2. Analysis of the Importance of Four Types of Features

We examine driver genes from a single feature type to discuss the best trait of features. According to the known literature, these 75 features can be divided into four types. Each feature helps determine whether it is a driver of cancer. By grouping related features, we can explain a small number of feature groups, each of which has a different biological explanation.

Our framework is analyzed in four ways with four indicators. In the accuracy index, it can be seen that the value of the mutational feature is higher than other features, and the same is true for other indicators. Among the four different model types, the performance of the logistic regression model is almost superior to other models, except for genomics and phenotypes features in the recall, which did not perform so well. However, due to the nature of data, neural networks are not considered suitable for this classification purpose. As shown in Figure 3, we find that mutational features are the best features for identifying cancer driver genes, superior to genetic, phenotypic, and genomic features.

### 3.3. Analysis of Top N Features

We detect driver genes from the top N features to discuss which feature combinations are the most effective. Different cancers may have different driver genes. Different driver genes may be caused by different characteristics. If a feature is important, its importance should not be diluted by adding another feature. The result of each algorithm is the average of 100 times. According to Table 1, we put the top N features into the classifier for learning to predict the importance of these features in cancer driver genes. It can be seen from Figure 4 that when the top 45 features are in a group, the four different algorithms are relatively high in the four indicators, except for in the accuracy index where logistic regression is not so obvious. The top 45 features included 21 mutation features, 16 epigenetics features, 7 genomics features, and 1 phenotype feature. Overall, the top 45 features are the most beneficial feature combinations.

We compare the top 45 features with the mutation features to discuss which is the better combination in Figure 5. The top 45 features have higher accuracy and F1-score than the mutation features, except in the neural network with an unsignificant difference. The top 45 features are significantly higher than the mutation features in precision and recall. On the whole, it is obvious that the top 45 features are the most effective feature combinations and superior to the mutational features. The top 45 features include not only mutational features but also three other types of features.

### 3.4. Comparison of Methods

Our framework analyzes the impact of multiple omics features on driver gene identification. The MutSigCV [38] method identifies cancer driver genes based on mutation characteristics, but those cancer driver genes with infrequent mutations are difficult to detect. OncodriveFML [39] uses the functional impact of gene mutations to reveal both coding and non-coding cancer drivers. GUST can predict a default value for TSG and OG, and GUST [40] software does not allow such threshold adjustment. DORGE [21] uses genetic and epigenetic genes to identify cancer driver genes. MutPanning [41] contains a database of driver genes for 28 different tumor types, as well as additional driver genes found through mutations in odd nucleotide contexts. However, our framework identifies cancer driver genes by integrating mutational, epigenetic, phenotypic, and genomic data.

Our framework is further compared with five existing cancer driver gene prediction algorithms using five precision metrics: recall, F1-score, specificity, accuracy, and precision. (Table 1). Our method ranks second in precision and accuracy indicators. However, the most obvious advantage is in recall of experimental results, with the highest performance (0.787), followed by DORGE (0.611), OncodriveFML (0.338), and MutPanning (0.318) (Table 1). The performance on specificity is the best, reaching (0.999) and also reaching the overall accuracy. In terms of index F1-score, our method reaches 0.765 higher than other methods. When compared on precision and accuracy, DORGE performs best, and ours comes in second.

**Table 1 genes-13-00716-t001:** Results of cancer gene identification.

Method	Recall	Specificity	F1-Score	Precision	Accuracy	Algorithms
Our framework	0.787	0.999	0.765	0.941	0.929	Logistic regression
MutSigCV [38]	0.137	0.998	0.731	0.905	0.888	Mutational Background
GUST [40]	0.206	0.994	0.713	0.838	0.894	Random forest
MutPanning [41]	0.318	0.994	0.729	0.880	0.907	Nucleotide context
DORGE [21]	0.611	0.997	0.723	0.966	0.948	Logistic regression with the elastic net
OncodriveFML [39]	0.338	0.915	0.685	0.367	0.841	Functional impact

### 3.5. Enrichment Analysis

For discussing the biological function of driver genes which are detected by our framework, we use the gene ontology (GO) enrichment analysis on the gene sets for the top N features. The driver genes detected by top N features under the logistic regression are analyzed by gene set enrichment analysis (GSEA) [42] (http://www.gsea-msigdb.org/gsea/msigdb/annotate.js (accessed on 1 August 2021)). The smallest *p*-value of each driver gene set is selected, and the *p*-value is transformed to be $-\log_{10}\left(p-value\right)$. It is clear to see that the driver gene set predicted under the top 45 features has the best enrichment, which is shown in Figure 6.

The driver genes detected by four machine learning algorithms with the top 45 features are analyzed by GSEA [42] and Enrichr [43] (https://maayanlab.cloud/Enrichr/ (accessed on 1 August 2021)) with gene ontology terms in order to study their biological function. The analyzed results for the driver gene set detected by logistic regression, neural networks, random forest, and XGBoost with GSEA and Enrichr gene ontology terms are presented in Appendix A, separately. The top 100 gene ontology terms with *p*-value < 0.05 are selected with GSEA for each driver gene set (Appendix A). The driver gene sets predicted by the four methods have 70 common GSEA gene ontology terms (Appendix A), such as GOBP_POSITIVE_REGULATION_OF_NUCLEOBASE_CONTAINING_COMPOUND_METABOLIC_PROCESS, GOBP_PROGRAMMED_CELL_DEATH, GOMF_TRANSCRIPTION_REGULATOR_ACTIVITY, GOCC_CHROMOSOME, and so on. The gene ontology terms with *p*-value < 0.05 are all selected in Enrichr for each driver gene set. The driver gene sets predicted by the four methods also have common gene ontology terms in Enrichr (Appendix A). Under the GO Biological Process, GO Cellular Component, and GO Molecular_ Function in Enrichr, the four methods predicted 49, 41, and 52 common terms, separately (Appendix A). Overall, most of common gene ontology terms are associated with cell death, cell differentiation, cell activation, immune system, and other biological processes, which are all important roles in the development of cancer.

### 3.6. Analysis of Driver Genes

Four machine learning algorithms are used in the framework and they all detect both known and unknown cancer genes. For these newly identified driver genes, we conduct a literature review to evaluate the evidence of association with cancer genes (Appendix A). Furthermore, the driver genes detected by the four algorithms not only include CGC genes but also non-CGC genes, which are considered as new driver genes (Table 2). Logistic regression predicts the most genes associated with more than five specific types of cancer. Logistic regression predicts 358 cancer drivers with high scores, 238 of which are found in known CGC cancer genes. Neural networks detect 291 genes as driver genes with high scores, 191 of which are discovered in known CGC cancer genes. The XGBoost method identifies 304 cancer driver genes above the threshold of the objective function, 174 of which are known CGC cancer genes. Random forest predicts 284 high-scoring genes, 184 of which are included in known CGC cancer genes. The overlap of cancer genes predicted by the four machine algorithms is shown in Figure 7. The genes within the CGC gene standard set are also listed in Appendix A. The cancers which driver genes are associated with are included in Appendix A. Overall, four machine learning algorithms can effectively detect both known and new cancer genes.

Some predicted new cancer genes which are not in CGC are also associated with cancer. Sox9 is a transcription factor that plays a key role in the development of many tissues. Sox9 has also been expressed in prostate cancer cell subsets and increased in recurrent hormone-refractory prostate cancer (PCa) [44]. KLF3 regulatory axis is involved in the development of lung cancer, suggesting a possible target for future lung cancer therapy strategies [45]. ACVR1B is linked to tumorigenesis through its interaction with activin-A [46]. RASA1 protein levels in RKO cells are much lower than in the other five colon cancer cell lines, indicating that miR-21 activated RAS signaling pathways by down-regulating RASA1 expression. It promotes cell proliferation, anti-apoptosis, and tumor cell development [47]. MLL3 and MLL4 are two of the most important players in enhancer regulation and cancer etiology. More and more research is being done on the role of enhancer failure in tissue-specific carcinogenesis [48].

## 4. Conclusions

This study emphasizes the identification of cancer driver genes by using four machine learning methods based on multi-omics features. Based on CGC and non-CGC genes, 75 features are divided into four categories and analyzed using KL divergence. We detect cancer driver genes in two different ways. One is to detect the driver gene from a single feature type, and the experimental results show that the mutation feature is the best. The other is to detect from the first N features. We find that the top 45 features are the most effective from the second analysis, and it is also outperforming only mutational features. These top 45 features do not merely contain mutation features, but also three other types of features. Thus, our framework not only considers the mutation characteristics in the patient’s gene but also considers other types of characteristics, such as genomic characteristics, epigenetic characteristics, and phenotypic characteristics. In addition, our method is superior to other methods such as DORGE, OncodriveFM, and MutPanning. Our method can better identify potential cancer driver genes.

However, our approach has some challenges. In future work, there are primarily two ways to improve. On the one hand, multiple omics features of paired genes, such as co-occurring or mutually exclusive pairs, can be integrated into this framework to find driver modules. It is beneficial to extract more specific information from different aspects. On the other hand, in clinical practice, we can discuss the significance of different types of features on precision medicine and personalized medicine.

## Figures and Tables

**Figure 1 genes-13-00716-f001:**
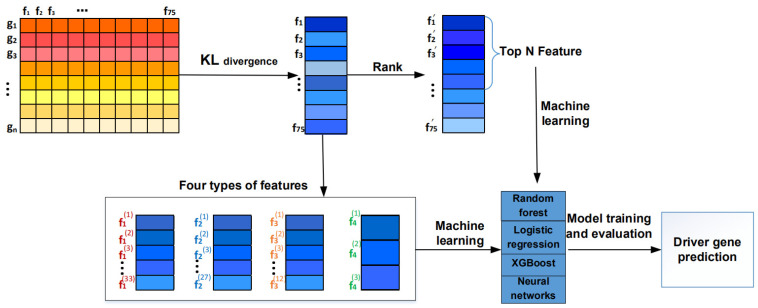
The framework of analyzing the influence of multiple omics features on driver gene identification.

**Figure 2 genes-13-00716-f002:**
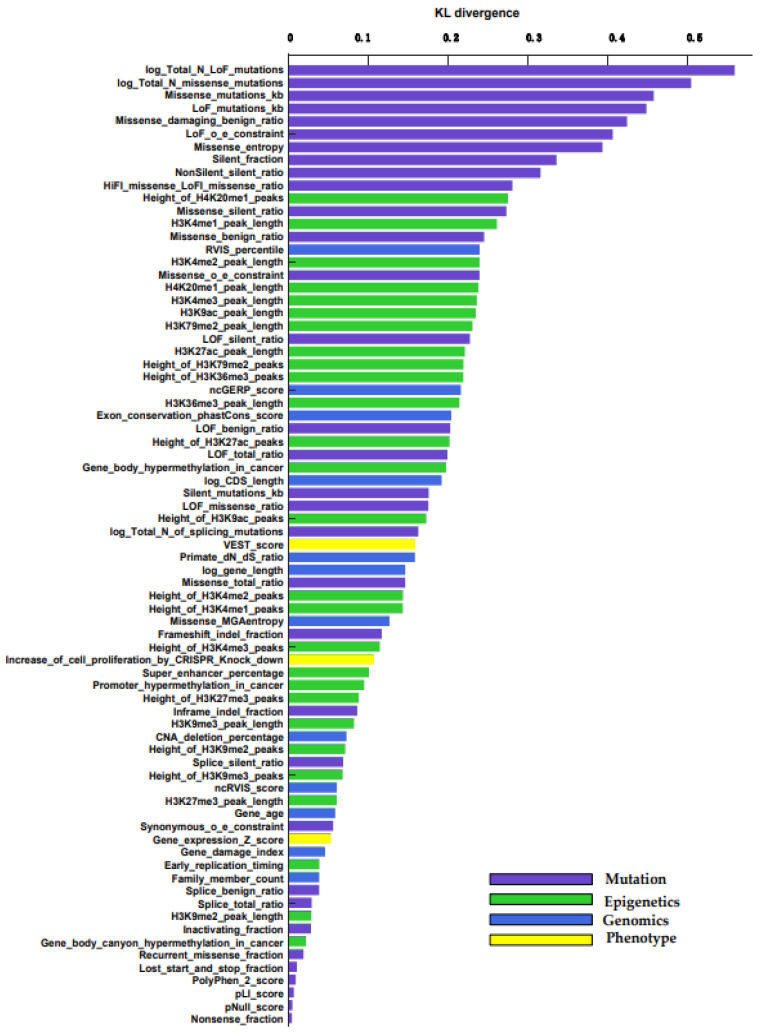
KL divergence of each feature is based on CGC genes and non-CGC genes.

**Figure 3 genes-13-00716-f003:**
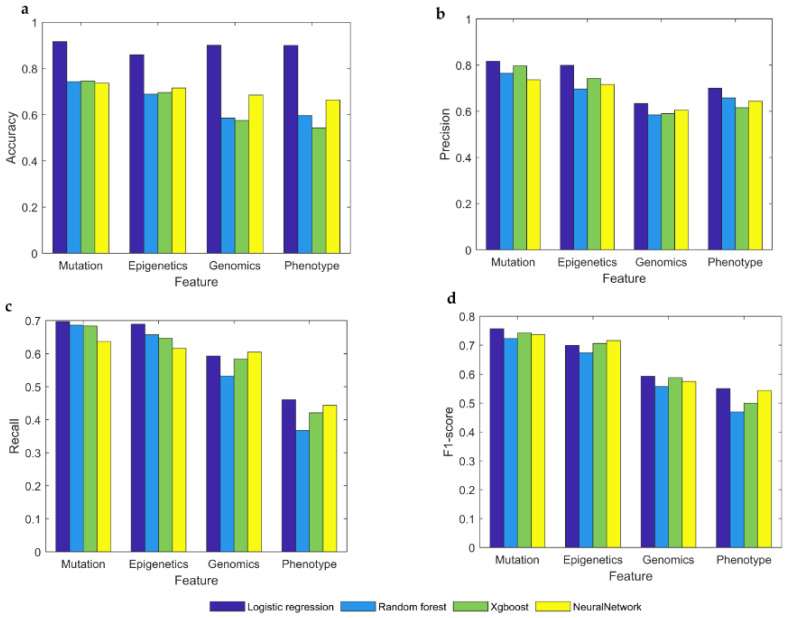
Detect driver genes from a single type of feature by four machine learning algorithms. (**a**–**d**) Compare four algorithms on accuracy, precision, recall, and F1-score. In each graph, the X-axis represents omics features. The Y-axis represents the value of accuracy, precision, recall, or F1-score, respectively.

**Figure 4 genes-13-00716-f004:**
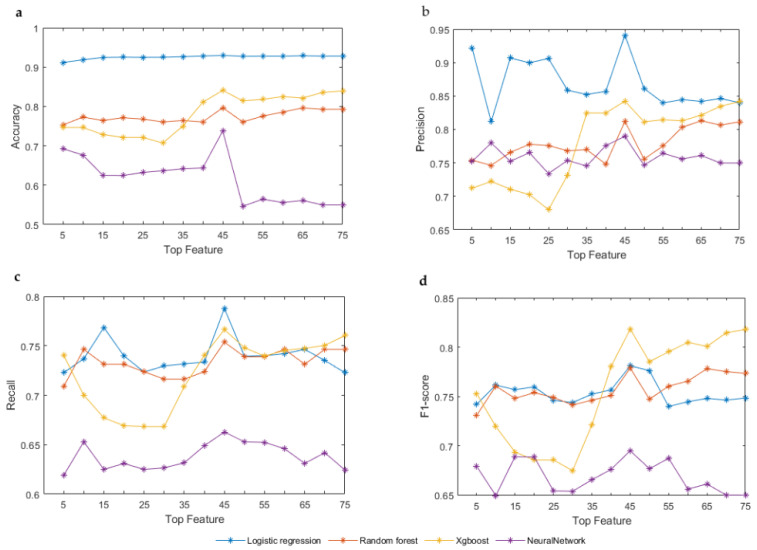
Detect driver genes from top N features by four machine learning algorithms. (**a**–**d**) Compare four algorithms on accuracy, precision, recall, and F1-score. In each graph, the X-axis represents the number of top N features. The Y-axis represents the value of accuracy, precision, recall, or F1-score, respectively.

**Figure 5 genes-13-00716-f005:**
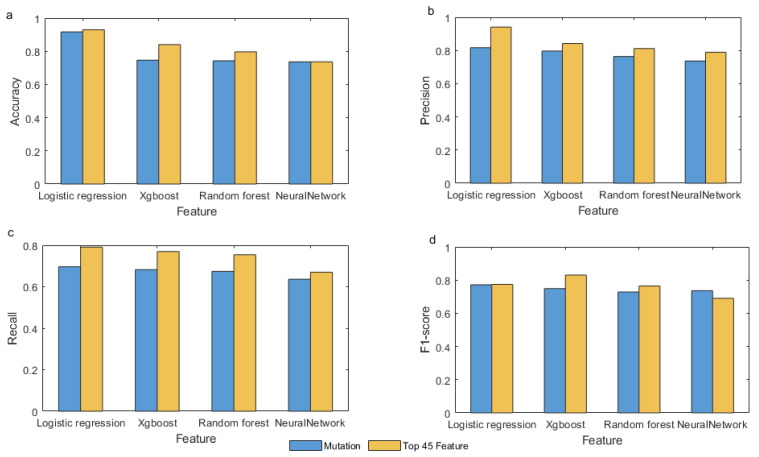
Compare the features of mutation types with the top 45 features, (**a**–**d**) in four different machine learning algorithms.

**Figure 6 genes-13-00716-f006:**
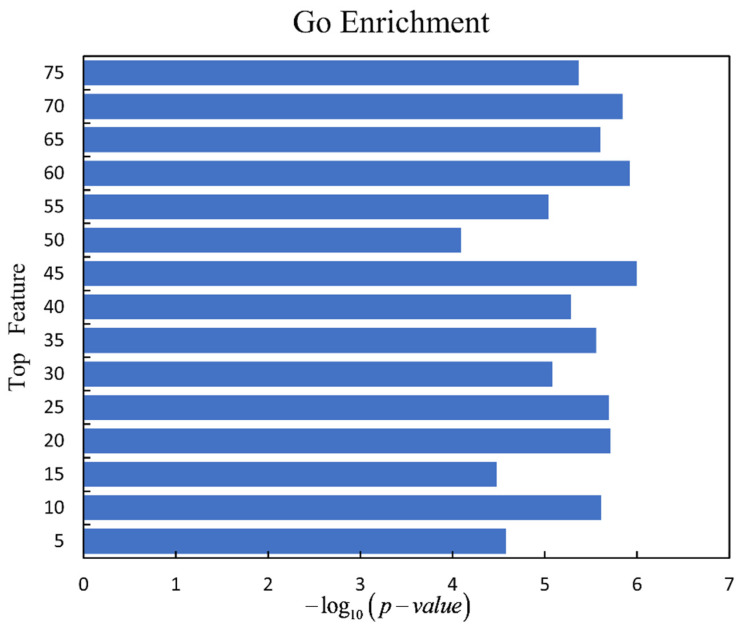
Enrichment of the driver gene sets predicted by the top N features using GSEA.

**Figure 7 genes-13-00716-f007:**
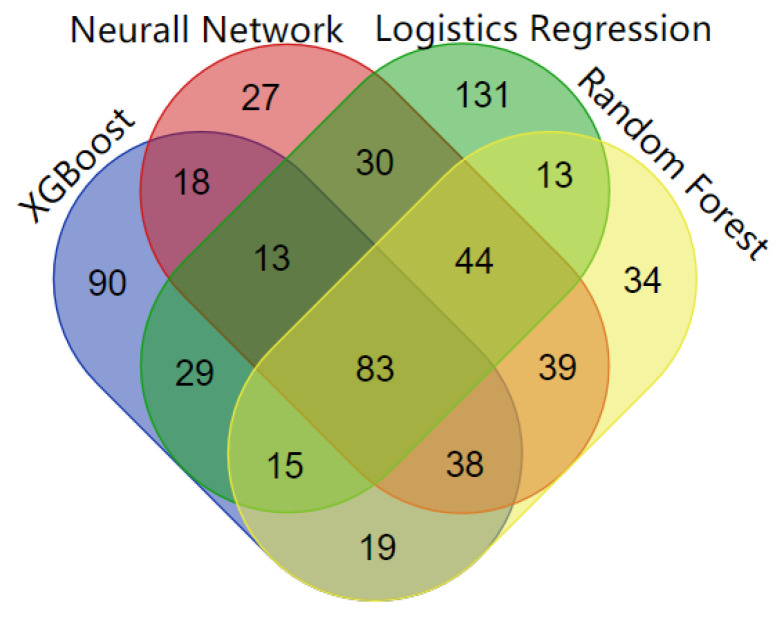
The overlap of cancer genes predicted by the four machine learning algorithms.

**Table 2 genes-13-00716-t002:** Number of driver genes detected by the four algorithms.

Algorithm	Total Gene	Non-CGC	CGC	CGC Genesin More than Five Cancer Types
XGBoost	304	130	174	11
Logistic Regression	358	122	236	22
Random Forest	284	99	185	8
Neural Network	291	101	190	13

## Data Availability

The datasets used in this study can be derived from the TCGA website (https://portal.gdc.cancer.gov/ (accessed on 1 August 2021)) and the COSMIC website (https://cancer.sanger.ac.uk/cosmic (accessed on 1 August 2021)).

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
