# Peer review of "Effects of Multi-Omics Characteristics on Identification of Driver Genes Using Machine Learning Algorithms"

_genes, 2022, doi:10.3390/genes13050716_

Round 1

Reviewer 1 Report

Dear Authors, thank you for this interesting piece of work. There some remarks:

  • please let native speaker review the manuscript; there are some minor issues
  • please explain all acronyms, abbreviations etc.
  • please consider adding a table with all discovered non-CGC genes not only selected ones; this small addidtion would make this work much more useful for clinicians 

Author Response

Response to reviewer 1

  1. Please let a native speaker review the manuscript. There are some minor issues.

Response: Thanks for the reviewer’s comments. We have seriously improved the English used in the entire manuscript accordingly and the polishing makes the manuscript more readable indeed. Since all of the changes are across the entire revised manuscript, we listed the details about corresponding changes in the latter part of the covering letter.

  1. Please explain all acronyms, abbreviations, etc.

Response: Thanks for the reviewer’s comments. We have added the full name for each abbreviation in the revised manuscript. For example, “deoxyribonucleic acid” is the full name for “DNA”, which is revised in Subsection1(paragraph 1 lines 11-12).

  1. Please consider adding a table with all discovered non-CGC genes not only selected ones; this small addition would make this work much more useful for clinicians.

Response: Thanks for the valuable suggestion. We added a table that contains both CGC and non-CGC genes detected by four algorithms, which is revised in subsection 3.6 (paragraph 1 lines 3-14). The cancers which driver genes are associated with are included in Supplementary Material 2.

Table 2. Number of driver genes detected by the four algorithms

Algorithm

Total gene

Non-CGC

CGC

CGC Genes in more

than five cancer types

XGBoost

304

130

174

11

Logistic Regression

358

122

236

22

Random Forest

284

99

185

8

Neural Network

291

101

190

13

Reviewer 2 Report

1. The feature set should be described in mor details. The reference on the feature set is shown only on the Figure 2.
2. The description of Kullback-Leibler distance, random forest method etc. can be ommited, because it is well known information that can be obtained from, say, Wikipedia.
3. The  Kullback-Leibler distance is not symmetrical. What genes were used as master and what genes were used as slave. May be symmetrized distance is more reliable. 
4. What is “co-length ding sequence”. Seems this is a Coding DNA Sequence (CDS).
5. What kind of the Artificial Neural Network was used? How many layers was used? What architecture of the ANN was used.
6. Figure 4. Almost all criteria have a strange peak on 45 features. Seems the authors provide testing only once and the feature set with 45 elements was biased. It is a good idea to do testing many times and to show mean and deviations of the criteria.
7. Seems it may be a good idea to add to the feature set GO categories. I think this kind of the feature will give about the best recognition.

Author Response

Responses to Reviewer 2

  1. The feature set should be described in more detail. The reference on the feature set is shown only in Figure 2.

Response: Thanks for the valuable suggestion. We have added a detailed description of the feature sets in the revised manuscript, like the reference (Lyu, Jie, et al., 2020) did, which is shown in section 3.1 (paragraph 1) and (Supplementary Table 1).

  1. The description of Kullback-Leibler distance, random forest method, etc. can be omitted because it is well-known information that can be obtained from, say, Wikipedia.

Response: Thank you very much for your advice. We redescribe these methods briefly, in the revised manuscript, which are more convenience for non-computer professionals.

  1. The Kullback-Leibler distance is not symmetrical. What genes were used as masters and what genes were used as a slave? Maybe symmetrized distance is more reliable.

Response: Thanks for the valuable suggestion. We are so sorry that we did not describe it clearly in the manuscript. Consider two different probability distributions  and , the average of  and is the final KL distance. And the final value is symmetric. The CGC genes group and non-CGC genes group are equivalent, and there is no distinction between these two groups. The details are described in section 2.2 (paragraph 2 lines 5-9).

  1. What is “co-length ding sequence”. Seems this is a Coding DNA Sequence (CDS).

Response: Thanks for the valuable suggestion. We are so sorry that we did not describe clearly and redescribed it in the revised manuscript, we have rewritten the Coding DNA Sequence (CDS), which is shown in Subsection 3.1(paragraph 2 line 4).

  1. What kind of Artificial Neural Network was used? How many layers were used? What architecture of the ANN was used.

Response: Thanks for the reviewer’s comments. The artificial neural network in this manuscript is a supervised learning error backpropagation algorithm. We used three layers, which include the input layer, output layer, and hidden layer. And a multilayer feedforward neural network is used, which is shown in section 2.3.4 (paragraph 1 lines 1-2 and paragraph 2 lines 1-2).

  1. Figure 4. Almost all criteria have a strange peak on 45 features. Seems the authors provide testing only once and the feature set with 45 elements was biased. It is a good idea to do testing many times and to show the mean and deviations of the criteria.

Response: Thanks for the valuable suggestion. We did not describe it clearly and re-described it in the revised manuscript. For the top N features, the result of each algorithm is the average of 100 times. We redescribe the experimental details, which are shown in section 3.3 (paragraph 1 lines 4-5).

  1. Seems it may be a good idea to add to the feature set GO categories. I think this kind of feature will give about the best recognition.

Response: Thanks for the valuable suggestion. We added some more descriptions to analyze the biological function of driver genes which are detected by our framework. We use the Gene Ontology (GO) enrichment analysis on the gene sets for the top N features. The driver genes detected by top N features under the logistic regression are analyzed by Gene Set Enrichment Analysis (GSEA) (http://www.gsea-msigdb.org/gsea/msigdb/annotate.js). We added Figure 6 to depict the Go enrichment pathway, which is shown in section 3.5 (paragraph 1 lines 1-9).

Reviewer 3 Report

Li and collaborators developed and explored a framework to modelling and to analyse the effects of multi-omics features on the identification of driver genes by four machine learning algorithms. The aim of developing this frameworks was to detect cancer driver genes in pan-cancer data, including 75 features among 19,636 genes.    

The four proposed algorithms were computationally tested using data from TCGA and COSMIC. The methods evaluated in the framework were:

  1. Logistic Regression
  2. Random Forest
  3. XGBoost
  4. Neural Network

Broad comments:

  • Line 181. It has to be clear whether 80% of the samples are cancer samples and non-cancer samples, as well as 20%; or how many percent of cancer samples and non-cancer samples there are in the training group and in the test group. In other words, the authors need to better explain how they created the training and test sets.

  • In figure 2, we clearly can see blocks of features importance-based (e.g. the first 10 features are from the mutation category). Is there some biological or statistical explanation for this result? Can the authors provide more feedback about this pattern?

  • Authors refered as Top N = 45 based in importance, in addition, when they evaluated (Figure 4) the 4 measurements (accuracy, Precision, Recall and F1-score) for the 45 features as a group, they found for Top features = 45 the maximum value for almost all measurements (except for logistic regression when evaluating accuracy). However, in figure 3 the showed that mutational features by themself were the best to identify cancer driver genes, and in line 255 they refered that Top = 45 is the only superior to mutational features. Then, maybe authors can make a sensitive analysis comparing both, mutational group and Top = 45 group, for the 4 measurements as showed in figure 4.

  • Authors did not describe any thing about figure 5.

  • Rewrite the first paragraph in section 3.4 (comparison of methods). Maybe to start with, what are authors going to do in this section, with which softwares they are going to compare their framework, why these software, what do these software, etc.

  • Authors skiped the section 3.5 in Results section.

  • Section 3.6. It looks like the framework detect very well driver genes than if the 4 algorithms are applied by themself. Is there a possibility that the authors could do a sensitivity analysis by decreasing the number of top features?

  • Section 3.6. Is the framework capable of detecting new driver genes? Namely, all the driver genes detected by the framework were already described in the literature or are new ones?

  • Section 3.6. Could the authors provide more information on the results at the biological level? For example, a table of how many of the driver genes detected by the framework are specific to a certain type of cancer, in which tissues, etc.

Specific suggestions:

  • Line 125. Reference 33 no correspond with Logistic regression.
  • Line 136. Reference 34 no correspond with Random forest.
  • Line 136. I suggestion change the expression: 'build a forest' to 'build a decision tree'.
  • Line 141. I did not understand the expression 'Each decision tree has its objective function, here we take 0.5...'. The objective function does not have a threshold, I never have seen this. Then, 0.5 is the threshold for decision tree. Please, rewrite this sentence.
  • Lines 145 and 157. Kernel function of Random forest and objective function of XGBoost need a reference. I looked for them in the cited articles and did not find the formulas. Sorry if I am wrong.
  • Line 160. It is preferable that authors use Artificial neural network rather than Neural network.
  • Line 162. There are many gradient descendent methods in Artificial neural network, like, batch gradient descent, stochastic gradient descent and mini-batch gradient descent. Why did the authors use the applied method?
  • Line 180. Please, change 'classified as cancer gene' to 'labeled as cancer genes'

Round 2

Reviewer 2 Report

All my suggestions are taken into account except one. I recommend to provide Go analysis. But this recommendation was provided only formally to get p-value. But to get the biologically relevant results it is necessary to show what kind of gene ontology terms are significant to understand what kind of genes were selected as important features to predict the driver mutations.
